# Integrating Smart Greenhouse Cover, Reduced Nitrogen Dose and Biostimulant Application as a Strategy for Sustainable Cultivation of Cherry Tomato

**DOI:** 10.3390/plants13030440

**Published:** 2024-02-02

**Authors:** Roberta Paradiso, Ida Di Mola, Lucia Ottaiano, Eugenio Cozzolino, Maria Eleonora Pelosi, Massimo Rippa, Pasquale Mormile, Mauro Mori

**Affiliations:** 1Department of Agricultural Sciences, University of Naples Federico II, 80055 Portici, Italy; rparadis@unina.it (R.P.); ida.dimola@unina.it (I.D.M.); mariaeleonora.pelosi@unina.it (M.E.P.); mori@unina.it (M.M.); 2Research Center for Cereal and Industrial Crops, Council for Agricultural Research and Economics (CREA), 81100 Caserta, Italy; eugenio.cozzolino@crea.gov.it; 3Institute of Applied Science and Intelligent System, National Research Council (CNR), 80078 Pozzuoli, Italy; m.rippa@isasi.cnr.it (M.R.); p.mormile@isasi.cnr.it (P.M.)

**Keywords:** *Lycopersicum esculentum* L., polyethylene, lycopene, soluble solids, texture, antioxidant capacity, ascorbic acid, phenols, carotenoids, pigments

## Abstract

Fruit yield and quality of greenhouse tomatoes are strongly influenced by light conditions and nitrogen (N) availability, however, the interaction between these factors is still unclear. We evaluated the effects on cherry tomatoes of two tunnel plastic covers with different optical properties and three N doses, also in combination with a biostimulant treatment. We compared a diffuse light film (Film1) and a conventional clear film (Film2), and three N levels, corresponding to 50% (N50), 75% (N75) and 100% (N100) of the optimal dose, with and without a microbial plus a protein hydrolysed biostimulant, compared to a non-treated control. The three experimental treatments significantly interacted on several yield and quality parameters. In control plants (untreated with biostimulants), the early yield was higher at reduced N doses compared to N100, with greater increments under the diffusive Film1 compared to the clear Film2 (+57.7% and +37.0% vs. +31.7% and +16.0%, in N50 and N75 respectively). Film1 boosted the total fruit production at all the N rates and with or without biostimulants, compared to Film2, with stronger effects under sub-optimal N (+29.4% in N50, +21.2% in N75, and +7.8% in N100, in plants untreated with biostimulant). Total yield decreased with decreasing N levels, while it always increased with the application of biostimulants, which counterbalanced the detrimental effects of N shortage. Quality traits were mainly affected by the cover film and the biostimulant treatment. The diffusive film increased the content of carotenoids, lycopene and total phenols compared to the clear one, and the biostimulants increased texture, soluble solids, phenols and ascorbic acid compared to the untreated control. It is worth noting that in plants fertilized at 75% of the reference N dose, the biostimulants determined higher yield than the N100 untreated control, under both the covers (+48% in Film1 and +20% in Film2). In conclusion, the diffusive film improved the fruit yield and quality of greenhouse tomatoes in the spring–summer period, presumably avoiding plant stress due to high-intensity direct light. Reduced N rates limited the plant productivity, however, the biostimulant application was effective in compensating for the detrimental effects of sub-optimal supply of N synthetic fertilizers.

## 1. Introduction

Sustainable agriculture is gaining global importance as a response to food and environmental challenges. In this respect, protected cultivation allows for the extension of the crop production season and the increase of produce yield and quality. However, greenhouse cultivation implies several practices with relevant environmental impact, including fertilization and energy use for automated operations and climate conditioning [1]. Specifically, soil nitrate pollution, primarily determined by agricultural fertilizers and animal manure, causes groundwater contamination, leading to eutrophication and harming aquatic ecosystems [2,3]. Hence, since proper plant nutrition is crucial to guarantee high-yield products and quality, adopting rational fertilization and monitoring its environmental effects is pivotal to addressing both food and environmental challenges.

Light quantity, as intensity and duration, and light quality, as wavelength composition, drive plant physiology and metabolisms along the entire life cycle, hence influencing the yield and quality of plant products. Indeed, plants use light as an energy source for photosynthesis, and also as a signal to regulate many other fundamental processes of growth and development in photomorphogenesis [4]. Global solar radiation consists of direct radiation, reaching a given surface straight from the sun, and diffuse radiation, reflected or scattered by atmospheric particles (e.g., water vapour, dust, pollutants), hence arriving from multiple directions [5]. Diffuse radiation determines a more uniform distribution of light in the vertical and horizontal space and penetrates deeper into the canopy, increasing the assimilation rate of lower and inner leaves [6]. Light can be a limiting factor in greenhouses, particularly in winter months, when solar radiation has a lower and more variable intensity. Indeed, light intensity is reduced compared to outside, due to the shading of the bearing structure and the incomplete transmissivity of the cover material (i.e., glass or plastics) [7], declining over time for wear and dust deposition [8]. Therefore, diffuse light represents an important portion of the solar radiation entering into greenhouses, especially in winter, and it is proven to be more effective in promoting light absorption, photosynthesis, and light use efficiency, finally enhancing the crop uniformity and productivity, compared to direct light [9].

Recently, innovative smart materials with specific optical features have been designed to modulate the light intensity, spectral composition, and distribution in protected cultivation [10]. For instance, diffusive covers spread light more uniformly compared to conventional clear covers, thanks to the inclusion into the matrix of interference pigments, gas microbubbles, or hollow glass microspheres [10], and their application is proven to improve plant growth [5]. Indeed, diffusive glass slabs and plastic films enhance the light distribution in the vertical profile in plants with upright habitus, increasing the light interception of lower and inner leaves (in winter), and mitigating the stress for excessive light and thermal levels in the upper and external leaves (in summer) [11]. Furthermore, an even light environment can improve the product quality in terms of nutrient and phytochemical composition [4]. Accordingly, covers with 90% transmittance and a minimum of 50% diffusivity are currently suggested [9].

Nitrogen fertilization influences plant nitrogen (N) metabolisms and overall mineral nutrition since N can interact with other nutrients in synergistic or antagonistic ways [12]. Furthermore, the nutrient uptake is influenced by light intensity and spectral composition [4], and by light direction as the angle of incidence of the beam on the leaf surface, which alters the light absorption and the assimilation rate [13]. Consequently, light diffusion, of both natural and artificial light, plays a key role in the plant nutrition of greenhouse crops.

Tomato (*Solanum lycopersicum* L.) is a highly popular fruit crop, grown and consumed across the entire world, and produced in both open fields and protected cultivation [14]. Recent assessments of the effects of tomato production in terms of human health and environmental impact of different cropping systems, including conventional and organic farming in open fields and greenhouses, highlighted the need for improvement of the agronomical practices (e.g., irrigation, fertilization) and the overall process sustainability (e.g., resources consumption and pollution) [14]. In this scenario, greenhouse cultivation offers the advantage of year-round production and higher yield and potentially allows for better management of irrigation and crop protection; however, the energy requirement for air conditioning and the high quantity of chemical fertilizers are critical issues for the greenhouse industry’s sustainability [14].

The relevance of the nutritional and nutraceutical value of fresh tomato in the human diet is universally recognized, and the valuable content of health-promoting phytochemicals has been well characterised [15]. Specifically, tomato is a good source of antioxidants and other bioactive compounds exerting a preventive effect against important chronic degenerative disorders. Indeed, fruits are rich in phenolics (phenolic acids and flavonoids), carotenoids (lycopene, and α and β carotene), vitamins (ascorbic acid and vitamin A) and glycoalkaloids (tomatine), with antioxidant, anti-mutagenic, anti-proliferative, anti-inflammatory and anti-atherogenic activity [15]. The bioavailability of phyto-constituents in tomato is generally not affected by routine cooking processes, making the fruit even more beneficial for human consumption [16]. However, as an intensive crop, tomato requires relevant inputs of chemicals and pesticides to guarantee optimal yields.

Several studies addressed the influence of cover features and N fertilization rate on greenhouse vegetables, though their interaction is still unclear. We set up an integrated strategy, implying: (i) the use of a diffusive film, to improve the light environment and therefore the plant growth and product quality, while increasing the natural light use efficiency; (ii) the reduction of the N dose, to optimize the N use efficiency while limiting the soil pollution; (iii) and the application of natural plant biostimulants to reduce the use of chemicals [17,18]. Plant biostimulants consist of natural substances and microorganisms able to enhance the efficiency of several physiological and molecular processes, such as carbon assimilation, mineral nutrition, and secondary metabolism (including synthesis of phytochemicals), improving the plant nutrition and tolerance to abiotic stress, and finally the yield and product quality [19]. They include non-microbial formulates, based on seaweed and plant extracts, microalgae, protein hydrolysates, and amino acids, as well as microorganisms, such as arbuscular mycorrhizal fungi and plant growth-promoting rhizobacteria (PGPR) [20]. Their application is widely spreading in horticulture to integrate or partially replace chemical fertilizers and appears as a promising solution to enhance crop productivity while reducing environmental pollution [21]. Among biostimulants, PGPR are receiving relevant attention.

Based on this premise, we hypothesized that a functional modification of the light environment combined with a rational N supply and a biostimulation treatment could guarantee high yield and quality of greenhouse tomatoes while reducing environmental impact. Hence, within a series of experiments aiming at evaluating the above-mentioned strategy, we assessed the influence of two plastic covers, a diffusive film (Film1) and a clear film (Film2), and three N regimes, corresponding to 50% (N50), 75% (N75) and 100% (N100) of the optimal theoretical dose, on cherry tomato grown in tunnels in spring–summer, also in combination with a biostimulant treatment. This treatment combined a microbial mix, including mycorrhizal fungi, rhizosphere bacteria, and *Trichoderma* spp., and a protein hydrolysed of alfalfa, containing 11% organic N, 25% organic C, and 70% total amino acids. We assumed that these products, applied together, could act by stimulating plant growth through the action of beneficial microorganisms, providing organic elements and compounds simultaneously to support this promoting effect while compensating for the reduced N supply.

## 2. Results

### 2.1. Air Temperature

Figure 1 shows the minimum and maximum temperature (per ten days) recorded in open air and in the tunnel with the two polyethylene covers, diffusive Film1 and clear Film2, during the experimental period. In the outdoor environment, the minimum temperature ranged from 7.6 °C (I decade of April) to 22.7 °C (II of July), and the maximum temperature from 15.4 °C (I decade of March) to 32.2 °C (II of July). In the two tunnels, the minimum temperature showed a similar trend, with the lowest and the highest values in the same decades, and a general tendency to slightly lower values under the clear Film2, but no relevant difference compared to outside. The maximum temperature in the tunnels reflected the time evolution of external values, with the lowest level at the beginning of March and at the second decade of April, for Film2 and Film1, respectively, and the highest in the middle of July. Both the cover materials determined an increase in the maximum temperature compared to open air, with a greater effect under Film1 in the first four decades of cultivation.

### 2.2. Early Production and Total Marketable Yield

Table 1 and Table 2 show the effects of the experimental factors (cover film, nitrogen fertilization, and biostimulant application) on the early marketable yield (first three harvests) and total marketable yield, and the main quality parameters of tomato fruits.

The analysis of variance revealed that all the tested treatments significantly influenced the production earliness and plant productivity (Table 1). Significant second-degree interactions, between plastic film and biostimulant (F × B), and third-degree interactions, among plastic film, N rate and biostimulant (F × N × B) were found in both these parameters, while N rate and biostimulant (N × B) interacted only on total yield.

Referring to the fruit quality, the plastic cover influenced only the dry matter percentage, while the effect of biostimulant on total soluble solids (TSS), texture, and dry matter at the last harvest (Table 1) was significant.

Among the experimental factors, plastic cover was the most impactful on the fruit colour and nutraceutical properties, influencing significantly all the considered parameters, except the blue-yellow component b* and the hydrophilic antioxidant activity HAA (Table 2). The effects of N fertilization were relevant to the green-red component a*, and the total content of phenols and N-Kjeldhal, while those of the biostimulant were to ABTS antioxidant activity, total phenols and ascorbic acid.

Significant second-degree interactions were the following: plastic film with nitrogen dose (F × N) and with biostimulant (F × B) on lycopene and N content and on lycopene, respectively. The N rate interacted with biostimulant treatment (N × B) on total phenols and N-Kjeldhal content.

The third-degree interaction Cover film × Nitrogen dose × Biostimulant application obtained in the early marketable production (sum of first 3 harvests, from 29 May to 16 June) is shown in Figure 2. In control plants (untreated with biostimulants), it was generally higher at the reduced N doses compared to N100, with greater increments under diffusive Film1 compared to the clear Film2 (+57.7% in N50 and +37.0% in N75, and +31.7% in N50 and +16.0% in N75, respectively), and it was unaffected by the cover material at the reference dose (N100).

The early production decreased in N100 in Film1, while it did not significantly change with the N rate under Film2 (Figure 2). The application of the biostimulants increased the early marketable yield under both the covers and at all the nitrogen fertilization levels, except N100 in Film2, where no differences emerged compared to the untreated control. The highest increment for biostimulant treatment was recorded in plants grown at N100 under Film1 (Figure 2).

The total marketable yield was also significantly influenced by the third-degree interaction Cover film × Nitrogen dose × Biostimulant application (Figure 3). In both control plants and plants treated with biostimulant, it was higher under Film1 compared to Film2, at every nitrogen regime. Particularly, the diffusive film increased the total yield compared to the conventional clear film, with stronger effects under sub-optimal N supply (i.e., +29.4% in N50, +21.2% in N75, and +7.8% in N100 in Film1, in plants untreated with biostimulant).

Total yield decreased at decreasing N level; however, the effect of limited N supply was lighter under the diffusive film (−36.8% and −10.8% at N50 and N75, compared to N100) than under the clear one (−51.6% and −23.7%, respectively).

Fruit yield always increased in the presence of biostimulants, which also counterbalanced the detrimental effects of N shortage (Figure 3). Specifically, the increase changed depending on the cover and the N regime (+63.9, +65.9 and +48.7 in Film1, and +27.9, +57.4 and +68.7 in Film2, at N100, N75 and N50, respectively, compared to the untreated controls).

The components of the early and total marketable yield in the different experimental treatments are shown in Table 3.

Regarding early marketable yield, in control plants untreated with biostimulants, the number of berries was higher under the diffusive Film1 compared to the clear Film2 at N50 (+44.9%), while it did not show a significant difference between the films at the higher doses. In control, both the reduced N doses increased the number of early fruits compared to N100, under both the cover films. The biostimulant treatment did not influence the number of berries, regardless of the cover material and the N fertilization, except for the treatment N100 under Film1, in which it determined a strong increase compared to the untreated control (+122%).

Regarding total marketable yield, the number of berries was generally higher in plants grown under Film1 compared to Film2 (Table 3). In control plants, it decreased at only N50 under both Film1 (−19.6%) and Film2 (−14.0% at N75 and −36.0% at N50), compared to N100. The application of biostimulants always determined the production of more fruits compared to the untreated control, with different increases depending on the film and the N dose (+14.0%, +42.7% and +53.6% at N50, N75 and N100 respectively, in Film1, and +39.1%, +37.3% and +22.4%, in Film2).

None of the treatments influenced the specific fruit weight (Table 3).

Table 4 shows the average effects of cover material, nitrogen level and biostimulant application on some quality features of cherry tomato berries at the third (16 June) and the last (12 July) harvest. Averaged on the other treatments, the cover material did not influence the fruit texture and soluble solids content, while the diffusive Film1 significantly reduced the dry matter percentage at both the harvests (−11.3% and −5.7%, respectively), compared to the clear one. The nitrogen dose did not affect any of the considered parameters, while the application of the biostimulants had a significant effect on all of them, except the dry matter percentage at the 3rd harvest, increasing the texture (+17.6%) and the total soluble solids (+6.9%), and reducing the dry matter percentage at the 7th harvest (−7.5%), compared to control.

The diffusive Film1 significantly reduced the brightness (L*) and the green-red component (a*), while promoting the synthesis of lycopene (Figure 3) and carotenoids (Table 5) of tomato fruits compared to the conventional Film2 (+24% on average). The nitrogen dose had a significant effect only on the component a*, which decreased in N50 compared to N100. The application of biostimulants did not alter the colour parameters or the carotenoid content.

The significant interactions found in lycopene content between the cover film and the nitrogen dose (F × N) and the cover film and the biostimulant application (F × B) (Table 2) are shown in Figure 4. Compared to N100, at N50 the lycopene content was higher under Film1 (+15.7%) and lower under Film2 (−24.3%) (Figure 4A). The application of biostimulants did not influence the lycopene compared to control within each cover, however, in fruits of treated plants, it determined higher values under Film1 compared to Film2 (+33%; Figure 4B).

The growth of tomato plants under the diffusive Film1 determined lower values of ABTS antioxidant activity and total ascorbic acid (TAA) (−28.2% and −61.2%, respectively) and higher values of total phenols (+4.3%), compared to the clear Film2 (Table 6). The effect of nitrogen fertilization on antioxidant activity and TAA was irrelevant, while the biostimulant increased the TAA content compared to the untreated control (+16.9%).

Nitrogen dose and biostimulant revealed a significant interaction on the total content of phenols in tomato fruits (N × B; Table 2).

Indeed, averaged on the cover films, the application of biostimulants determined different effects on total phenols depending on the N rate, with significant increases at N50 and N100 and no relevant effect at N75, compared to the untreated controls (Figure 5).

The significant interactions of nitrogen dose with both the cover film and the biostimulant application on the N-Kjeldhal content in tomato fruits (F × N and N × B; Table 2) are reported in Figure 6. Compared to N100, the N-Kjeldhal significantly decreased only at N75 under the light-diffusive Film1 and at both the reduced doses under the clear Film2 (Figure 6A). The biostimulants influenced the content of N-Kjeldhal differently at the different N rates, determining an increase at N50 and a decrease at N75 compared to controls and no significant effect at N100 (Figure 6B).

## 3. Discussion

Modulation of light intensity via diffusive covers is a promising strategy to improve the light environment in greenhouse horticulture in both Mediterranean and temperate climates, making light distribution more uniform while preventing stress conditions for excessive or insufficient light intensity in the canopy [5]. However, while the influence of diffuse light on crop performance is gaining increasing attention, knowledge of the effects on plant nutrition is still scarce. Similarly, the side effects on soil microbiota (including beneficial microorganisms) have been barely investigated.

We investigated the effects of diffusive light and reduced nitrogen supplies and the efficacy of a biostimulant treatment on cherry tomatoes grown in a tunnel in spring–summer in Southern Italy, with the overall objective to improve the sustainability of the production process. We compared two PE covers—a diffusive film (58% diffusivity, 90% total and 30% UV-B transmissivity) and a conventional clear film (85% transmittivity)—and three N doses corresponding to 50%, 75% and 100% of the optimal quantity (equal to 160 kg ha^−1^), with or without the application of Lifestrong VAM (containing beneficial rhizobacteria and *Trichoderma*) and Biolifestrong (from the enzymatic hydrolysis of alfalfa).

The growth of tomatoes under the diffusive film boosted the fruit production in the first three harvests (from 29 May to 16 June) at the reduced N doses, and the total yield (from 29 May to 12 July) at all the N rates, compared to the conventional clear film, through a higher number of fruits, in both control and biostimulant-treated plants. Hence, diffuse light increased the early production compared to direct light under lower N availability. This result could be due to the nutritional stress from N shortage which prompted plants to anticipate flowering and fruit setting [22], which were better sustained by the more favourable diffuse-light environment. Consistently, also under the clear film, the early production was slightly higher in N50 and N75 than in N100, although the differences did not reach statistical significance. This evidence on production earliness is commercially relevant, as the cherry tomato hybrid ‘Sakura’ exhibits a continuous production of berries, giving multiple harvests, and the early produce reaches a higher market price than the ordinary one in a standard period, being highly appreciated by consumers. Our results on the influence of diffuse light on productive performance agree with other evidence in Mediterranean greenhouses (Almeria, southern Spain), in which a plastic diffusive cover enhanced the yield of tomatoes in the spring–summer period (+3.2% more than a commercial thermal film), due to the increase of both the number of fruits per plant and the average fruit weight [23]. Similarly, Dueck et al. [24] observed increases in production of 10% in tomatoes under diffusive glass in The Netherlands.

In the long term, the sub-optimal N supply limited plant productivity, reducing the number of fruits and slightly limiting the specific fruit weight, regardless of the film type and also in the presence of biostimulants. This evidence confirms that the theoretical quantity of 160 kg ha^−1^ is (or is close to) the optimal N dose for tomato and that lower supplies compromise the fruit yield [25]. However, it is worth noting that the application of the biostimulants in plants fertilized at 75% of the reference N dose gave a higher yield than the N100 untreated control, under both the covers (+48% in Film1 and +20% in Film2), more than compensating for the sub-optimal N fertilization.

Averaged on the other treatments, the diffuse light reduced the dry matter percentage compared to the direct light, differently from previous experiments, in which three diffuse-light glasshouses (45%, 62%, and 71% haze) did not alter the dry matter percentage of tomato fruits compared to a standard horticultural glass [26].

The diffuse light promoted the synthesis of lycopene, total carotenoids, and total phenols in tomato fruits. This effect could be ascribed to the role of these antioxidant compounds in the plant reaction to the light stress induced by UV-B solar radiation (280–315 nm), partially transmitted by the tested diffusive film. An extensive body of literature documents the harmful effects of ultraviolet radiation (UVA and UVB) reaching the Earth’s surface in higher plants and the related metabolic responses (for example, see [27]). Particularly, UV-B-induced cellular damage and processes mediated by the photoreceptor UV resistant locus8 (UVR8), including the biosynthesis of carotenoids (comprising lycopene) and phenols [28], acting as UV-absorbing screen and limiting penetration of UVB into the plant tissues, are reported for several horticultural crops [29]. Furthermore, diffuse light is also known to promote the synthesis of phytochemicals and antioxidants in tomato [30]. However, in five cultivars grown in six independent greenhouses, including two standard transparent and two diffusive PE films, and two diffusive polycarbonate sheets, Ahmadi et al. [31] found a relevant variability in the content of carotenoids and polyphenolic compounds and genotype-specific responses to light environment in their accumulation.

Interestingly, in our experiment, the ABTS antioxidant activity and the total ascorbic acid were much higher in fruits obtained under the clear Film2 (+40% and +145%, respectively, compared to the diffusive Film1). In this respect, it is conceivable that the high intensity occurring in the South of Italy in the spring–summer period, transmitted as direct light under the clear tunnel, determined light and thermal stress at the plant level, which also stimulated the antioxidant response [32].

Nitrogen fertilization did not affect most of the tested parameters of fruit quality (i.e., texture, total soluble solids, dry matter percentage, antioxidant activity, and total ascorbic acid), while the halved dose increased accumulation of total phenols compared to the optimal dose, under both the covers. This could also be a part of the response to the above-mentioned stress for N shortage, implying early fruitification [33]. Differently, the response to N availability in terms of lycopene changed under the covers, with the highest content under the diffusive film and the lowest under the clear film at N50. It is conceivable that even though the N shortage could have acted as a stressor, triggering the synthesis of antioxidants (i.e., phenols and lycopene), this was made possible by diffuse light, while it was impeded by direct light for lycopene since its synthesis is inhibited by high temperature [34] and radiation [35].

Under both covers, the application of the microbial biostimulants Lifestrong Vam L followed by the weed extract Biolifestrong increased the yield, texture, soluble solids, and ascorbic acid content of tomato fruits compared to the untreated control. Lifestrong Vam L contains the bacteria *Bacillus subtilis* and *Pseudomonas fluorescens*, known as phosphorus solubilizer, and *Azospirillum brasilense*, acting as asymbiotic atmospheric N-fixing, and *Trichoderma* endophytic fungi, improving soil nutrient availability and functionally modulating the root growth [20]. Biolifestrong is based on alfalfa hydrolysed, providing amino acids and peptides as well as non-protein components ready to be taken up by plants [20]. Positive responses in terms of plant growth have been observed in tomatoes inoculated with microbial biostimulants, including both PGPR and *Trichoderma* fungi [36]. He et al. [37] reported that three species of *Bacillus* and one of *Pseudomonas*, inoculated either individually or in co-culture, promoted the growth and the uptake of N-depleted fertilizer of tomato plants, and co-inoculation with more microorganisms with complementary modes of action exerted an additive effect of growth promotion. Recently, Cirillo et al. [38] showed how the simultaneous application of *Trichoderma afroharzianum* and *Azotobacter chroococcum* enhanced yield and alleviated the effects of combined water–nitrogen stress in tomato. Furthermore, the use of a plant-based biostimulant (sugar cane molasses with yeast extract by yeast autolysis) improved plant performances and fruit quality in tomato-grown plastic tunnels in Southern Italy at summer elevated temperatures [39].

Several mechanisms have been described for plant growth and nutrient uptake stimulation by microbial inoculants, including the following: (i) asymbiotic atmospheric N fixation, (ii) solubilization of nutrients, (iii) sequestering of iron through siderophores production, and (iv) synthesis of volatile organic compounds (VOCs) [20]. Likewise, the phytostimulation action of *Trichoderma* can involve direct and indirect effects on plant metabolism, physiology, and morphology, including the release in the rhizosphere of secondary metabolites with hormonal activity (i.e., auxin-like compounds), small peptides and volatile organic compounds (VOCs), stimulating the root development (total length, number, and branching of roots) hence improving the root system architecture, and the increase of availability of soil elements, including macro-(P) and micro-(Fe, Mn, Zn) nutrients [40]. Hence, in our experiment, in the presence of a mixed culture including different microbial species and plant hydrolysed, the better performance of treated tomato plants under different light environments and N supplies could have involved several of the listed processes. However, it is worth noting that the efficacy of the biostimulant treatment on plant growth and the effects on plant metabolism depend on specific plant–microbe interactions and environmental conditions [20]. For instance, in our experiment, the biostimulants interacted with the light environment in the lycopene content, with a higher value under Film1 compared to Film2, and with the N rate in total phenols, which increased under N50 and N100 supply compared to non-treated control, confirming the influence of microorganisms on the antioxidant machinery.

## 4. Materials and Methods

### 4.1. Plant Material and Crop Management

The experiment was carried out during the spring–summer of 2023, from 6 March to 12 July, under two twin tunnels (same features in terms of shape, dimensions, and volume) at the Department of Agricultural Sciences of the University of Naples Federico II, in Portici (Naples, Italy; 40°49′ N, 14°20′ E; 70 m a.s.l.).

Plants of tomato (*Lycopersicum esculentum* L.), bunch cherry hybrid ‘Sakura’ (Ensa Zaden, Tarquinia, Viterbo, Italy), were grown in pots (0.38 m^2^) on sandy soil (USDA classification). The main physical and chemical properties of the soil are detailed in Table 7.

The hybrid is characterised by early production, small round fruits (18/22 g), and red colour [41]. Seedlings were transplanted on 6 March, at the planting density of 5 plants per m^2^, and harvest was staggered across 7 dates: 29 May, 6, 16, 23 and 28 June and 7 and 12 July.

The air temperature inside the tunnels was monitored hourly by sensors (Vantage Pro2, Davis Instruments).

Irrigation was based on the restitution of water losses per evapotranspiration, estimated according to the Hargreaves and Samani formula [42].

Crop protection consisted of 4 treatments with Spinosad (active principle: extract of *Saccharopolyspora spinosa* bacterial culture) to control *Noctuidae lepidoptera* and *Tuta absoluta*.

### 4.2. Plastic Covers, Nitrogen Rates, and Biostimulant Application

The two tunnels were covered with polyethylene (PE) thermal films (thickness 150 μm), with an anti-drip effect. Film1 (trade name Sunsaver Diff, manufacturer Ginegar Plastic Products and distribution Polyeur Srl, Benevento, Italy) was a light diffusive film, with 58% diffusivity, 87% thermicity, 30% transmissivity in the UV-B waveband (280-315 nm), 90% total (direct + diffused) transmittivity in photosynthetically active radiation (PAR). Film2 (trade name Lirsalux, Lirsa Srl, Ottaviano, Naples, Italy) was in clear plastic, with 75% thermicity, no transmission in the UV-B range, and 85% total transmittivity of PAR.

Nitrogen fertilization was performed at three rates, corresponding to 50% (N50), 75% (N75) and 100% (N100) of the optimal N dose, equal to 160 kg of N per hectare, calculated according to the Campania Region Fertilization Plan [23]. Ammonium nitrate (N content 26%) was used as fertilizer, applied in 3 interventions on a monthly basis, starting about 30 days after transplanting (DAT).

The biostimulant treatment was performed using two commercial products, Lifestrong Vam L and Biolifestrong, according to the strategy suggested by the firm (Fertilidea Srl, Pompei, Naples, Italy [43]). Lifestrong Vam L is a soil organic amendant (liquid formulate), including mycorrhizal fungi 100 spores/g, rhizosphere bacteria 3 × 10^^8^ UFC/g (*Bacillus subtilis*, *Pseudomonas fluorescens*, *Azospirillum brasilense*) and *Trichoderma* spp. 1 × 10^^8^ UFC/g. Biolifestrong (powder formulate) is obtained with the enzymatic hydrolysis of alfalfa (*Medicago sativa*) and contains 11% organic N, 25% organic C, and 70% total amino acids. The products are commonly used for treatments of seeds and plants and, according to the manufacturer’s claim, improve root and leaf growth and crop productivity. The biostimulant application included 5 treatments, one at transplanting with Lifestrong Vam L (immersion of the seedling roots for about an hour, at the dose suggested by the firm), and 4 during the growing cycle with Biolifestrong by foliar spraying, at 25 DAT, then approximately bi-weekly, at the dose suggested by the firm (150 g/100 L).

The timing of nitrogen supply and biostimulant application and the dates of the harvests are reported in Table 8.

### 4.3. Time Distribution of Production and Total Yield, and Product Quality

At each harvest, the number of marketable berries and the total fresh weight were measured, and yield was expressed in kg per m^2^. The sum of the first three harvests was considered the early yield, and the sum of all harvests was the total yield.

At the third and the last harvest, a representative sample of berries for each pot was taken and dried at 70 °C until a constant weight, to calculate the dry matter content.

#### 4.3.1. Texture

On the same samples, texture was measured with a digital penetrometer (T.R. Turoni s.r.l., Forlì, Italy) with a tip of 8 mm, and expressed in kg m^−^^2^.

#### 4.3.2. Total Soluble Solids

On fresh tomato juice, the total soluble solids were determined with a portable digital refractometer (model DBR 35, Sinergica Soluzioni s.r.l., Pescara, Italy) and expressed as °Brix.

#### 4.3.3. Colour Parameters

In samples collected at the last harvest, on two sides of the equatorial zone of 5 berries per treatment, the colour parameters were measured with a Minolta CR-300 portable colorimeter (Minolta Camera Co., Tokyo, Japan). Colour measurements were expressed through the CIELAB (Commission international de l’eclairage) parameters L* (brightness), a* (green-red component), and b* (blue-yellow component).

#### 4.3.4. Lycopene

The lycopene content was determined according to the method of Sadler et al. [44]. The samples of tomato (2.5 g) were mixed with 50 mL of a mixture of n-hexane:acetone:ethanol (2:1:1) at 0.5% BHT (2,6-di-tert-butyl-4metyl-phenol). The mixture was agitated continuously for 30 min; 1 mL of the organic layer was carefully drawn out. This solution was collected into a 15 mL standard flask and made up to the mark with n-hexane. The absorbance was read at 472 nm using spectrophotometer and the total lycopene content was expressed as mg lycopene 100 g^−^^1^ fw.

#### 4.3.5. Carotenoids

A sample of 1 g of fresh fruit extracted with ammoniacal acetone was used to determine carotenoid content spectrophotometrically according to the method of Lichtenthaler and Wellburn [45]. Results were expressed in mg g^−^^1^ fresh weight (fw).

#### 4.3.6. Antioxidant Activity

The hydrophilic antioxidant activity (HAA) and ABTS antioxidant activity (ABTS AA) were measured spectrophotometrically on extracts from freeze-dried fruits (200 mg), in methanol and distilled water, according to the procedure of Fogliano et al. [46] and Re et al. [47], respectively. The extracts’ absorbance was read at 505 nm (HAA) and 734 nm (ABTS) and expressed as mmol ascorbic acid 100 g^−^^1^ dw and mmol of Trolox, respectively.

#### 4.3.7. Total Phenols

The content of total phenols was determined in methanolic extract through the Folin-Ciocalteu method [48], using gallic acid as a standard. An aliquot of 100 mL of the supernatant was mixed with 500 mL of Folin-Ciocalteau’s reagent (Sigma-Aldrich Inc., Milan, Italy) and 400 mL of 7.5% sodium carbonate/water (*w*/*v*). The absorbance was read after 30 min at 765 nm by a UV-Vis spectrophotometer, and the total phenols content was expressed as mg gallic acid 100 g^−^^1^ dw.

#### 4.3.8. Total Ascorbic Acid

The total ascorbic acid (TAA) was determined spectrophotometrically according to the protocol of Kampfenkel et al. [49], as the sum of ascorbic acid (ASA) and dehydroascorbate (DHA) acid and expressed as mg ascorbic acid 100 g^−^^1^ fw. The absorbance of the solution was read at 525 nm.

#### 4.3.9. Nitrogen

The nitrogen content of fruits was determined on dried samples by the Kjeldhal method [50], and the results were expressed as g kg^−^^1^.

### 4.4. Experimental Design and Statistical Analysis

The experiment was performed according to a split–split plot design, considering the 2 tunnel covers (Films) as the main experimental factor, and both the 3 N rates and the 2 biostimulant treatments as sub-factors. Under each film, treatments were applied in 3 replicates (3 pots per combination N rates × Biostimulant treatment), with 18 pots in total (3 N rates × 2 Biostimulant × 3 replicates).

All data were subjected to a 3-way analysis of variance (ANOVA), using the SPSS software package (SPSS version 22.0, Chicago, IL, USA). The means were compared using Tukey’s multiple range test at *p* = 0.05.

## 5. Conclusions

In conclusion, the diffusive cover improved the fruit yield and quality in greenhouse tomatoes grown in the spring–summer period in Southern Italy, presumably avoiding light stress conditions for high-intensity direct light at the plant level, typical in Mediterranean greenhouses, and leading to a higher light use efficiency. Reduced N rates limited the plant productivity, however, the biostimulant treatment compensated for the lower use of synthetic fertilizers. In fact, under both covers, the biostimulants allowed crops in N50 to reach a similar yield to the N75 untreated control, and in N75 to overcome the yield of N100 control. This last effect was more evident under diffusive film, with a yield increase that was more than double that observed under the clear film. Therefore, our results suggest that integrating diffusive light film, 75% of the optimal N dose, and biostimulant application could be a valuable strategy for sustainable tomato cultivation. In addition, the moderate nutritional stress induced by N shortage, in the improved diffuse light environment and in the presence of biostimulation, increases early tomato fruit production, with positive outcomes in terms of farm profits.

## Figures and Tables

**Figure 1 plants-13-00440-f001:**
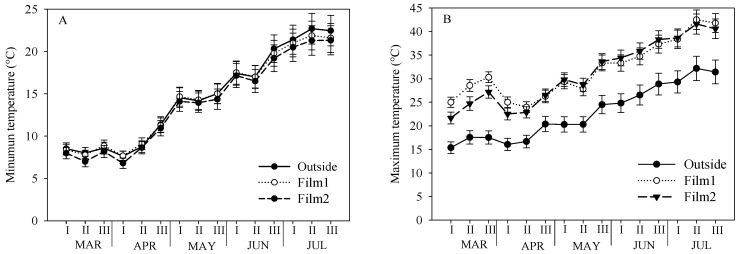
Ten-day values of minimum (**A**) and maximum (**B**) temperature recorded in open air (Outside) and in the tunnel with polyethylene covers, diffusive Film1 and clear Film2, during the experiment on cherry tomato (6 March–12 July 2023). Mean value ± Standard Error.

**Figure 2 plants-13-00440-f002:**
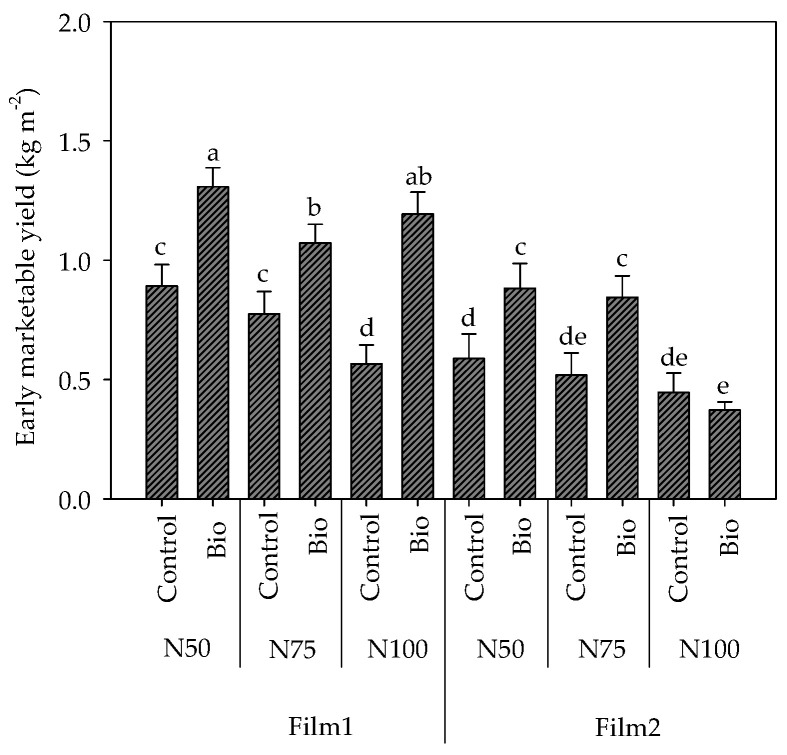
Interactions Cover film × Nitrogen dose × Biostimulant application (F × N × B) on the early marketable yield (as sum of the first 3 harvests) in cherry tomatoes grown in tunnel in spring–summer period. Different letters indicate significant differences. Mean value ± Standard error (n = 3).

**Figure 3 plants-13-00440-f003:**
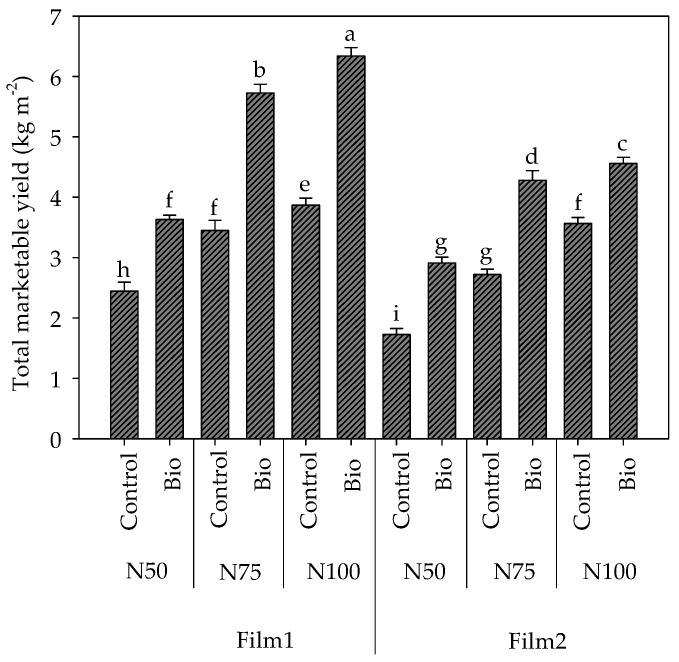
Interactions Cover film × Nitrogen dose × Biostimulant application (F × N × B) on the total marketable yield in cherry tomato grown in tunnel in spring–summer period. Different letters indicate significant differences. Mean value ± Standard error (n = 3).

**Figure 4 plants-13-00440-f004:**
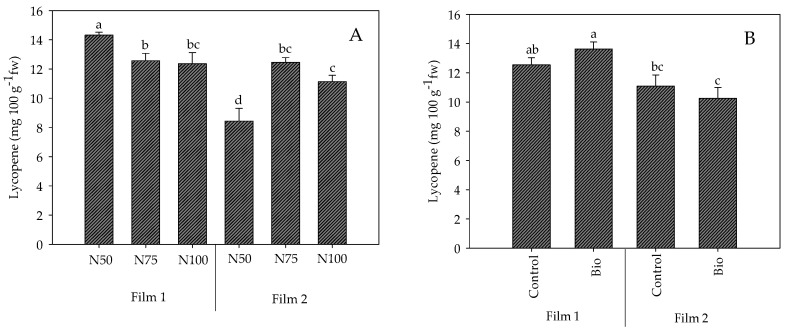
Interactions Cover film × Nitrogen dose (F × N) (**A**) and Cover film × Biostimulant application (F × B) (**B**) on lycopene content in fruits of cherry tomato grown in tunnel in spring–summer period. Mean value ± Standard error (n = 3). Different letters indicate significant differences.

**Figure 5 plants-13-00440-f005:**
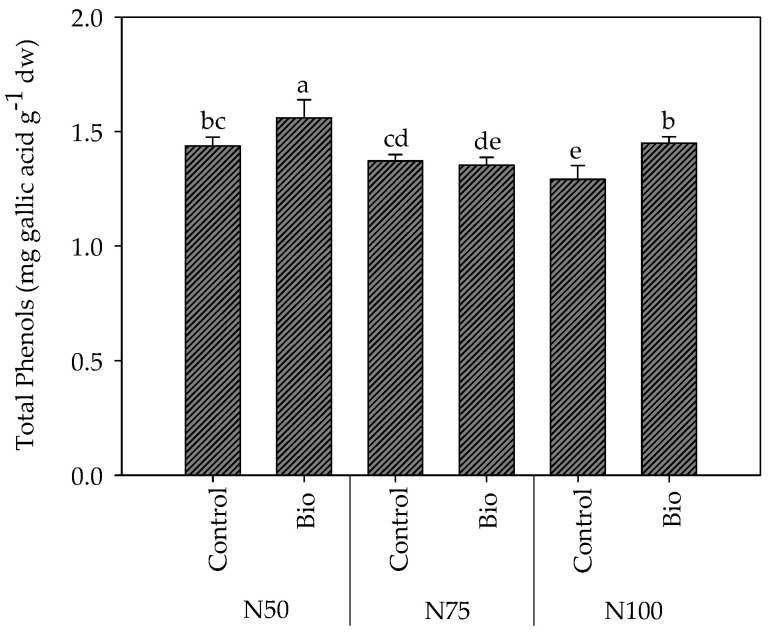
Interaction of Nitrogen dose × Biostimulant application on total phenols content in fruits of cherry tomato grown in tunnel in spring–summer period. Mean value ± Standard error (n = 3). Different letters indicate significant differences.

**Figure 6 plants-13-00440-f006:**
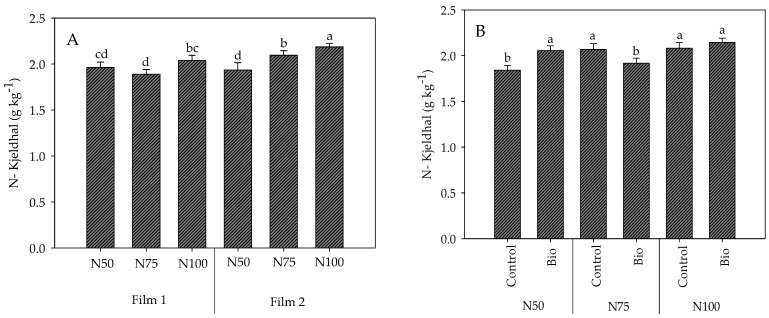
Interaction Cover film × Nitrogen dose (**A**) and of Nitrogen dose × Biostimulant application (**B**) on N-Kjeldhal content in fruits of cherry tomato grown in tunnel in spring–summer period. Mean value ± Standard error (n = 3). Different letters indicate significant differences.

**Table 1 plants-13-00440-t001:** Analysis of variance on early and total marketable yield and their components, and on texture, total soluble solids (TSS), and dry matter percentage of tomato fruits as affected by the different experimental treatments: significance of main factors and interactions.

	Early Marketable Yield			Total Marketable Yield			Texture	TSS	Dry Matter %	Dry Matter %
	kg m^−2^	n. Fruits	g Fruit^−1^	kg m^−2^	n. Fruits	g Fruit^−1^			3rd Harvest	7th Harvest
Film (F)	0.001	0.001	0.001	0.001	0.001	0.05	ns	ns	0.001	0.05
N Fertilization (N)	0.001	0.001	ns	0.001	0.001	0.001	ns	ns	ns	ns
Biostimulant (B)	0.001	0.01	0.001	0.001	0.001	0.001	0.001	0.001	ns	0.01
F × N	ns	0.05	ns	ns	ns	ns	ns	ns	ns	ns
F × B	0.05	ns	ns	0.001	0.01	ns	ns	ns	ns	ns
N × B	ns	ns	0.01	0.001	0.05	ns	ns	ns	ns	ns
F × N × B	0.05	0.01	ns	0.001	0.01	ns	ns	ns	ns	ns

ns: non-significant; differences at *p* ≤ 0.05, 0.01 and 0.001 (Tukey’s HSD test).

**Table 2 plants-13-00440-t002:** Analysis of variance of colour parameters (L*: brightness; a*: green-red component; and b*: blue-yellow component), lycopene and carotenoids content, hydrophilic (HAA) and ABTS antioxidant activity (ABTS AA), total phenols, total ascorbic acid (TAA), and N-Kjeldhal, as affected by the different experimental treatments: significance of main factors and interactions.

	L*	a*	b*	Lycopene	Carotenoids	HAA	ABTS AA	Total Phenols	TAA	N-Kjeldhal
Film (F)	0.01	0.001	ns	0.001	0.001	ns	0.001	0.01	0.001	0.001
N fertilization (N)	ns	0.05	ns	ns	ns	ns	ns	0.001	ns	0.001
Biostimulant (B)	ns	ns	ns	ns	ns	ns	0.05	0.001	0.001	ns
F × N	ns	ns	ns	0.001	ns	ns	ns	ns	ns	0.05
F × B	ns	ns	ns	0.05	ns	ns	ns	ns	ns	ns
N × B	ns	ns	ns	ns	ns	ns	ns	0.01	ns	0.001
F × N × B	ns	ns	ns	ns	ns	ns	ns	ns	ns	ns

ns: non-significant; differences at *p* ≤ 0.05, 0.01 and 0.001 (Tukey’s HSD test).

**Table 3 plants-13-00440-t003:** Interactions Cover film × Nitrogen dose × Biostimulant application on the components of the early (as sum of the first 3 harvests) and total marketable yield in cherry tomato grown in tunnel in spring–summer period. Mean value ± Standard error (n = 3).

			Early Marketable Yield		Total Marketable Yield	
			n. Fruits m^−2^	g Fruit^−1^	n. Fruits m^−2^	g Fruit^−1^
Film1	N50	Control	52.6 ± 4.6 a	18.9 ± 1.6	115.3 ± 6.1 e	22.3 ± 0.9
	N50	Bio	53.6 ± 4.6 a	25.6 ± 1.1	144.2 ± 4.3 d	25.7 ± 0.4
	N75	Control	37.6 ± 3.0 cd	21.6 ± 1.6	138.7 ± 4.2 d	24.7 ± 0.8
	N75	Bio	40.3 ± 4.1 cd	27.1 ± 1.6	197.9 ± 7.4 b	28.8 ± 0.7
	N100	Control	22.9 ± 5.5 e	26.2 ± 2.2	143.4 ± 5.2 d	26.7 ± 0.8
	N100	Bio	50.8 ± 3.2 ab	24.7 ± 1.2	220.3 ± 8.6 a	28.5 ± 0.7
Film2	N50	Control	36.3 ± 4.7 cd	16.8 ± 2.0	85.5 ± 5.5 f	24.4 ± 0.7
	N50	Bio	43.4 ± 4.7 bc	21.6 ± 1.3	118.9 ± 4.5 e	24.7 ± 0.6
	N75	Control	34.7 ± 7.9 d	18.2 ± 1.4	115.0 ± 7.9 e	23.9 ± 1.0
	N75	Bio	36.3 ± 6.0 cd	24.3 ± 0.8	157.9 ± 8.4 c	26.7 ± 0.5
	N100	Control	20.3 ± 3.9 e	21.0 ± 1.2	133.7 ± 2.9 d	26.1 ± 0.2
	N100	Bio	18.9 ± 2.0 e	20.4 ± 2.2	163.7 ± 4.3 c	26.9 ± 0.4

Within each column, different letters indicate significant differences.

**Table 4 plants-13-00440-t004:** Effect of cover film, nitrogen dose and biostimulant application on berry quality parameters in cherry tomato grown in tunnel in spring–summer period. Mean value ± Standard error (n = 3).

	Texture kg cm^−2^	Total Soluble Solids (TSS, °Brix)	Dry Matter % 3rd Harvest	Dry Matter % 7th Harvest
Film1	0.74 ± 0.02	9.66 ± 0.13	9.4 ± 0.1 b	9.9 ± 0.2 b
Film2	0.74 ± 0.02	9.84 ± 0.13	10.6 ± 0.2 a	10.5 ± 0.3 a
N50	0.75 ± 0.02	9.57 ± 0.15	10.1 ± 0.2	10.1 ± 0.2
N75	0.73 ± 0.02	9.87 ± 0.16	9.9 ± 0.2	10.0 ± 0.3
N100	0.74 ± 0.02	9.79 ± 0.16	9.9 ± 0.1	10.4 ± 0.3
Control	0.68 ± 0.01 b	9.42 ± 0.13 b	10.2 ± 0.2	10.6 ± 0.3 a
Bio	0.80 ± 0.02 a	10.07 ± 0.10 a	9.8 ± 0.2	9.8 ± 0.2 b

Within each column, different letters indicate significant differences.

**Table 5 plants-13-00440-t005:** Effect of cover film, nitrogen dose and biostimulant application on CIELAB colour parameters, and carotenoid content of fruits grown in tunnel in spring–summer period. Mean value ± Standard error (n = 3).

	L*	a*	b*	Carotenoids mg g^−1^ fw
Film1	30.8 ± 0.3 b	20.0 ± 0.4 b	13.7 ± 0.3	0.136 ± 0.006 a
Film2	32.1 ± 0.3 a	22.7 ± 0.5 a	14.1 ± 0.2	0.110 ± 0.005 b
N50	31.5 ± 0.4	20.2 ± 0.7 b	13.7 ± 03	0.123 ± 0.007
N75	31.2 ± 0.4	21.9 ± 0.6 ab	13.7 ± 0.2	0.124 ± 0.007
N100	31.7 ± 0.4	22.0 ± 0.6 a	14.2 ± 0.3	0.122 ± 0.009
Control	31.8 ± 0.3	21.2 ± 0.5	14.0 ± 0.2	0.120 ± 0.005
Bio	31.1 ± 0.3	21.5 ± 0.5	13.7 ± 0.2	0.126 ± 0.007

L*, brightness: from black (0) to white (100); a*: from green (−60) to red (+60); b*: from blue (−60) to yellow (+60). Within each column, different letters indicate significant differences.

**Table 6 plants-13-00440-t006:** Effect of cover film, nitrogen dose and biostimulant application on Hydrophilic Antioxidant Activity (HAA), ABTS antioxidant activity (ABTS AA), and total ascorbic acid (TAA) in fruits of cherry tomato grown in tunnel in spring–summer period. Mean value ± Standard error (n = 3).

	HAA mmol ascorbic acid equ. 100 g^−1^ dw	ABTS AA mmol Trolox equ. 100 g^−1^ dw	TAA mg 100 g^−1^ fw	Total Phenols mg gallic acid g^−1^ dw
Film1	8.83 ± 0.13	8.49 ± 0.26 b	17.6 ± 1.2 b	1.44 ± 0.04 a
Film2	8.71 ± 0.16	11.83 ± 0.22 a	45.3 ± 1.2 a	1.38 ± 0.04 b
N50	9.03 ± 0.13	10.19 ± 0.46	32.9 ± 4.9	1.50 ± 0.05 a
N75	8.71 ± 0.20	10.03 ± 0.74	29.6 ± 4.3	1.36 ± 0.05 b
N100	8.59 ± 0.19	10.26 ± 0.52	31.8 ± 4.0	1.37 ± 0.05 b
Control	8.85 ± 0.14	9.81 ± 0.45	29.0 ± 3.6 b	1.37 ± 0.04 b
Bio	8.69 ± 0.15	10.51 ± 0.48	33.9 ± 3.4 a	1.45 ± 0.04 a

Within each column, different letters indicate significant differences.

**Table 7 plants-13-00440-t007:** Physical and chemical properties of the soil used in the experiment for cultivation of cherry tomato in plastic tunnels.

Soil Properties	Measure Units	Mean Values
Sand	%	91.0
Silt	%	4.5
Clay	%	4.5
N—total (Kjeldahl method)	%	0.101
P_2_O_5_ (Olsen method)	ppm	253.0
K_2_O (Tetraphenylborate method)	ppm	490.0
Organic matter (Bichromate method)	%	2.5
pH		7.4

**Table 8 plants-13-00440-t008:** Timing of the N fertilization, biostimulant application, and harvests, in cherry tomato grown in tunnel in spring–summer period, expressed as days after transplant (DAT).

Agronomic Practices	DAT						
Nitrogen fertilization	30	60	90				
Biostimulant application	1	26	51	76	101		
Harvest time	85	93	103	110	115	124	129

## Data Availability

Data are contained within the article.

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
