# Peer review of "Integrating Smart Greenhouse Cover, Reduced Nitrogen Dose and Biostimulant Application as a Strategy for Sustainable Cultivation of Cherry Tomato"

_plants, 2024, doi:10.3390/plants13030440_

Round 1
Reviewer 1 Report
Comments and Suggestions for Authors
The manuscript presents a study of the smart greenhouse cover, reduced nitrogen dose and biostimulant application on the yield and quality of greenhouse tomato. Considering that their interaction on the yield and quality of greenhouse tomato is still unclear, I think that the manuscript presents interesting findings and most of the research objectives have been achieved. Overall, the manuscript is worthy of publication to a large extent. However, I have several concerns that should be addressed before the paper could be published.
1. In the Abstract part, it is better for the authors show the novelty of the present study in a clearer way by supplementing the related contents, such as “their interaction on the yield and quality of greenhouse tomato is still unclear”. Moreover, it is better for the authors to supplement the conclusion by integrating the effect of the N level, the application of biostimulant and the PE cover films. In addition, it is also better to put forward the significance of the present study at a larger scale based the former results or conclusions at the end of the Abstract.
2. In the Result part, please check the term “per decade” (Page 3 of 17, Line 121). In addition, please the revise the format of the Table 2.
3. In the Discussion part, please supplement the conclusion by integrating the effect of the N level, the application of biostimulant and the PE cover films. In addition, it is also better to put forward the significance of the present study based the former results or conclusions at a larger scale at the end of the part.
Author Response
We thank the reviewer for his valuable revision. Below we report brief responses to the comments. In the manuscript the new text is highlighted in green color and the deleted parts as track changes.
- In the Abstract part, it is better for the authors show the novelty of the present study in a clearer way by supplementing the related contents, such as “their interaction on the yield and quality of greenhouse tomato is still unclear”. Moreover, it is better for the authors to supplement the conclusion by integrating the effect of the N level, the application of biostimulant and the PE cover films. In addition, it is also better to put forward the significance of the present study at a larger scale based the former results or conclusions at the end of the Abstract.
We thank the reviewer for the suggestion. The abstract has been modified according to the requests of all the reviewers (see track changes).
- In the Result part, please check the term “per decade” (Page 3 of 17, Line 121). In addition, please the revise the format of the Table 2.
We replaced “per decade” with “per ten-days” (line 120 pag.3). As regards the Table 2, we are not sure to understand your request, since the format of Table 2 is the same of Table 1.
- In the Discussion part, please supplement the conclusion by integrating the effect of the N level, the application of biostimulant and the PE cover films. In addition, it is also better to put forward the significance of the present study based the former results or conclusions at a larger scale at the end of the part.
We thank the reviewer for these comments. The Discussion has been revised according to the suggestions of all the reviewers and a paragraph of Conclusions has been added at the end of the manuscript.

Reviewer 2 Report
Comments and Suggestions for Authors
Comments and Suggestions for Authors
This study was conducted on the effects of two plastic covers with different optical properties and reduced N doses, in combination with the application of a microbial biostimulant, on cherry tomatoes grown in the tunnel in spring-summer.
The Idea of the paper is good, but the English of the text needs a little revision, especially the result section, I would suggest the authors have it checked preferably to avoid any mistakes. Also, there are some comments, please see them:
Abstract
The abstract is not well structured, without impressive data about results and weak conclusions. Please highlight your result based on percentage (Bring some values in the abstract, some number values like how much changed by %). The concluding remarks of the abstract are not well-written. Please add quantitative findings and method limitations at the end of the abstract section.
Keywords: should be three to ten, please remove one of them.
Introduction
The introduction has been well written, but since the tomato is a particular crop that is cultivated in both open-field (farm) and green-house, it would be better if could bring some information about disadvantages of open-field cultivation (Paragraph 5, lines 80-92), it could highlight the novelty of the current research, So, I would suggest that in two or three sentences explain the advantages and disadvantages of these two cultivation systems. Here is a published work that you can use it to improve this gap. https://doi.org/10.3390/agronomy13030916
Results
Please check Table 2 (some letters are extra)
Please check the significant letters in Figure 4, (Control, VAM, N75).
Discussion
I would suggest that after paragraph 1, add two or three sentences regarding the knowledge limitation about the role of plastic covers and nutrient management.
Line 330: add reference(s).
Line 338: … from previous experiments ….: after this reference (24), bring a brief result about this study and compare it with your research.
Line 344: what is your mean, …. [as example, see 25]????
Line 352: For a sentence with one reference, it’s better to bring the reference at the end of the sentence.
The main problem of discussion is that the authors didn’t compare their results with other studies appropriately.
Material and methods
It is better to bring the soil characteristics in the form of a table (lines 420-422).
Conclusion
Based on the Journal template, conclusion should bring after the material and method section. The conclusion section is missing in your paper. Please add it!
Comments on the Quality of English LanguageEnglish of the text needs a little revision, especially in the result section.
Author Response
We thank the reviewer for his valuable revision. Below we report brief responses to the comments. In the manuscript the new text is highlighted in red color and the deleted parts as track changes.
Abstract
The abstract is not well structured, without impressive data about results and weak conclusions. Please highlight your result based on percentage (Bring some values in the abstract, some number values like how much changed by %). The concluding remarks of the abstract are not well-written. Please add quantitative findings and method limitations at the end of the abstract section.
The abstract has been improved according to these comments and the requests of other reviewers.
Keywords: should be three to ten, please remove one of them.
One keyword has been deleted.
Introduction
The introduction has been well written, but since the tomato is a particular crop that is cultivated in both open-field (farm) and green-house, it would be better if could bring some information about disadvantages of open-field cultivation (Paragraph 5, lines 80-92), it could highlight the novelty of the current research, So, I would suggest that in two or three sentences explain the advantages and disadvantages of these two cultivation systems. Here is a published work that you can use it to improve this gap. https://doi.org/10.3390/agronomy13030916
We thank the reviewer for the suggestion. Some lines about tomato cultivation in open field and greenhouse have been added in the current version.
Results
Please check Table 2 (some letters are extra)
We are not sure to understand well this request; we assume the referee refers to the letters L*, a*, and b*, hence we added the explanation in the table capture.
Please check the significant letters in Figure 4, (Control, VAM, N75).
We checked the significance letters and we confirm that they are correct since value of W in Tukey Test is 0.07.
Discussion
I would suggest that after paragraph 1, add two or three sentences regarding the knowledge limitation about the role of plastic covers and nutrient management.
We thank the reviewer for this suggestion, which helped us to improve the Discussion. Some sentences about the limited knowledge on the interaction between light environment in greenhouse due to the plastic cover features and nutrient management have been added in the revised version.
Line 330: add reference(s).
In this sentence we just summarized the effects of N shortage, clarifying that, despite the promotion of flowering and fruit setting (determining the increase of yield at the first harvests – namely “early production”), in the long term (the whole cultivation cycle), the sub-optimal N supply limited the plant productivity. This is why we did not add a reference.
Line 338: … from previous experiments ….: after this reference (24), bring a brief result about this study and compare it with your research.
We thank the reviewer for this comment. The discussion about the comparison of our results with findings of other authors has been added.
Line 344: what is your mean, …. [as example, see 25]????
At the beginning of the sentence we wrote “An extensive literature documents the harmful effects of the ultraviolet radiation”, however we cited only one reference, hence we wrote “as an example (of this literature), see 25”.
Line 352: For a sentence with one reference, it’s better to bring the reference at the end of the sentence.
In previous papers, from both our team and other authors, published in MDPI Journal, we found that when the first author is reported in the text, the relative reference number is placed just after. We found many examples for this, hence we thought it was a general rule.
The main problem of discussion is that the authors didn’t compare their results with other studies appropriately.
We thank the reviewer for highlighting this weakness of the paper. Discussion has been improved by adding a comparison with other data on the topic.
Material and methods
It is better to bring the soil characteristics in the form of a table (lines 420-422).
We thank for this suggestion. The table has been added in M&M as Table 7.
Conclusion
Based on the Journal template, conclusion should bring after the material and method section. The conclusion section is missing in your paper. Please add it!
Although the Authors Guideline of Plants specifies that the Conclusion section is not mandatory, we moved our conclusions (previously at the end of Discussion) to a new Conclusion paragraph.

Reviewer 3 Report
Comments and Suggestions for Authors
The submitted manuscript to PLANTS-MDPI entitled “Integrating Smart Greenhouse Cover, Reduced Nitrogen Dose and Biostimulant Application as a Strategy for Sustainable Cultivation of Cherry Tomato” is interesting to investigate. BUT, following are the comments that need to be addressed:
What is the rationale behind using these three aspects/approaches together? Which one of these three was the best strategy to improve the tomato yield?
What was the reason of choosing these specific biostimulants? Were there any preliminary experiments with various biostimulants?
Abstract is poorly written. Please improve it extensively. It lacks the main results, conclusion, and future perspectives.
Please try to explain the results with fold-changes or percentage-changes when comparing the two treatments.
Tables: Why there are some significance letters missing for some treatments?
Why did the authors mention *reduced* nitrogen dose in title, however the maximum yield was attained at N100?
The conclusion section needs extensive revision. Please mention the most significant parameters that support your hypothesis and conclusion.
Author Response
We thank the reviewer for his valuable revision. Below we report brief responses to the comments. In the manuscript the new text is highlighted in blue color and the deleted parts as track changes.
What is the rationale behind using these three aspects/approaches together? Which one of these three was the best strategy to improve the tomato yield?
The importance of light environment and N availability on plant growth and productivity is well known, since these factors have been deeply investigated individually. To date, two needs have still to be satisfied: the increasing food demand linked to the increasing world population and a more sustainable crop cultivation, reducing the environmental impact of food production. Literature reports that yield decreases under decreasing N supply and increases under both diffuse light conditions and biostimulant application, however the interactions among the three factors is still unclear, hence we investigated if and to which extend one or both the last two conditions could compensate the detrimental effect of sub-optimal N doses. This rational is described in the last two paragraphs on the Introduction.
As regard the question “Which one of these three was the best strategy to improve the tomato yield?”, our results suggest that an integrated approach, involving light diffusive film, 75% of the theoretical optimal N dose and biostimulant application, could be an effective strategy for sustainable tomato cultivation. We added this sentence also in Conclusion.
What was the reason of choosing these specific biostimulants? Were there any preliminary experiments with various biostimulants?
Our biostimulant treatment combines a microbial mix, containing mycorrhizal fungi, rhizosphere bacteria, and Trichoderma spp., and a protein hydrolysed of alfalfa, containing 11% organic N, 25% organic C, and 70% total amino acids. We assumed that these products, applied together, could act by stimulating the plant growth through the action of beneficial microorganisms, providing at the same time organic elements and compounds to compensate the reduced N supply. This rational has been clarified in the revised version of the paper.
Abstract is poorly written. Please improve it extensively. It lacks the main results, conclusion, and future perspectives.
We thank the reviewer for highlighting this weakness of the manuscript. The abstract has been revised with track changes according to the requests of all the reviewers.
Please try to explain the results with fold-changes or percentage-changes when comparing the two treatments.
The increase and decrease have been added as percentage where they were missing in Abstract and Results.
Tables: Why there are some significance letters missing for some treatments?
In the tables the significance letters are missing only if the treatments (Film, N fertilization and Biostimulant) had a non-significant effect on the tested parameters (please, see the summary of the significance in tables 1 and 2). In addition, below each table we reported the sentence “In each column, different letters indicate significant differences” for better explaining the presence or absence of letters.
Why did the authors mention *reduced* nitrogen dose in title, however the maximum yield was attained at N100?
In the title we wanted to highlight that our approach aimed at integrating the three agricultural practices to develop a more sustainable cultivation strategy of cherry tomato. We did not mean to reveal the results or communicate any statement. This was the intent of the title, despite the fact that, the application of biostimulants in plants under the two reduced N doses (N75 and N50) actually compensated the yield reduction due to the N shortage compared to the optimal N dose (N100).
The conclusion section needs extensive revision. Please mention the most significant parameters that support your hypothesis and conclusion.
We thank the reviewer for this suggestion, which helped us to improve our manuscript. We revised the Conclusion paragraph according to the requests of all the reviewers (see track changes).

Reviewer 4 Report
Comments and Suggestions for Authors
The paper shows data on tomato production under cover in tunnel. The effect of three elements is studied; plastic cover, nitrogen dose and biostimulant use.
The data is organized differently depending on the table figure and the authors do not properly describe the data arrangement. Data is arranged in 3rd, 7th, early, spring-summer period and total production. One must assume that “total production” is also referred to as “spring-summer period” which comprises all months of the experiment. However, the authors are not clear about this. For example, In line 222 marketable “early production” and “total yield” is used but “Early production” and “total marketable yield” is used in figure 3. Authors should avoid changing the reference to the data as they are fragmenting it unequally in the different figures and tables. It would be advisable to increase the reading of the manuscript by homogenizing these terms, for example by using “Early production” and “total production”. Why the data is arranged in early and total production? Was the late production not significant? The authors should define clearly the ranges of data they analyzed specially if these ranges change in each figure/table.
Error bars should be added to figure 1.
The sentence seems to be cropped in line 187. Previous page finishes with “The N rate interacted with biostimulant treatment (N x B) on total phenols and N-Kjeldhalcontent.” and the next page begins with “degree interaction Cover film x Nitrogen dose x Biostimulant application obtained in the early production (first 3 harvests, from May 29 to June 16) is shown in Figure 2.”
The description of the experimental design should be improved. In line 125 and 417 it is mention that there are 2 tunnels but it is not mentioned how these two tunnels were used. Were the treatments and controls in the same tunnel during the repetitions, Were all repetitions made on the same tunnel? In line 519 shortly describes that “split-split design” was used not considering the two different covers. In line 521 is mention that all treatments were replicated 3 times (3 pots per replicate, 36 pots in total). Were these three replicates independent experiments, in different tunnels? How were the treatment combinations made? Related to these, the authors should indicate the N data shown in each table and figure as they do for table 3 “Mean value ± Standard error (n=3).”
The complexity of the data comparison should be improved by carefully describing why and how the data is arranged. The authors should also improve the description of the experimental design in the material and methods section.
Author Response
We thank the reviewer for his valuable revision. Below we report brief responses to the comments. In the manuscript the new text is highlighted in violet color and the deleted parts as track changes.
We thank the reviewer for highlighting the lack of clarity regards the crop productivity. The early production is the sum of the first three harvests, while the total yield is the sum of all harvests (including the first three) at the end of the cultivation cycle. This is now specified in the manuscript. In addition, we added “marketable” where it was missing, as suggested.
As regards the questions “Why the data is arranged in early and total production? Was the late production not significant?”, data were presented as early and total production because this tomato hybrid is characterized by a continuous production of berries (requiring multiple harvests) and the early production is appreciated by consumers and can be sold on the market at higher price than the ordinary production in the standard period. A sentence highlighting this aspect has been added in Discussion.
Error bars should be added to figure 1.
Done.
The sentence seems to be cropped in line 187. Previous page finishes with “The N rate interacted with biostimulant treatment (N x B) on total phenols and N-Kjeldhal content.” and the next page begins with “degree interaction Cover film x Nitrogen dose x Biostimulant application obtained in the early production (first 3 harvests, from May 29 to June 16) is shown in Figure 2.”
We apologize for the oversight. The formatting problem has been solved and the first part of the sentence is not hidden by Table 2 in the current version of the manuscript.
The description of the experimental design should be improved. In line 125 and 417 it is mention that there are 2 tunnels but it is not mentioned how these two tunnels were used. Were the treatments and controls in the same tunnel during the repetitions, Were all repetitions made on the same tunnel? In line 519 shortly describes that “split-split design” was used not considering the two different covers. In line 521 is mention that all treatments were replicated 3 times (3 pots per replicate, 36 pots in total). Were these three replicates independent experiments, in different tunnels? How were the treatment combinations made? Related to these, the authors should indicate the N data shown in each table and figure as they do for table 3 “Mean value ± Standard error (n=3).”
We thank the reviewer for highlighting the lack of clarity in the description of the experimental design, helping us to improve this part of the paper. In the paragraph “4.1. Plant Material and Crop Management” it is now clarified that the facilities were two twin tunnels (same features in terms of shape, dimensions, and volume), with two different plastic covers, a light diffusive film (Film1) and a conventional clear film (Film2). The tunnel plastic cover (Film) was the main experimental factor and, under each Film, the other two treatments (3 N rates and 2 biostimulant treatments) were applied in 3 replicates (3 pots per combination N rates x biostimulant treatment), hence under each Film there were 18 pots in total (3 N rates x 2 Biostimulant x 3 replicates).
Finally, we wrote that “The experiment was performed according to a split-split plot design, considering the 2 covers as the main factor, and the 3 N rates and the 2 biostimulant treatments as sub-factors”.
As regards the last comment, we added “Mean value ± Standard error (n=3)” in the capture of Figg. 2 and 3.
The complexity of the data comparison should be improved by carefully describing why and how the data is arranged. The authors should also improve the description of the experimental design in the material and methods section.
The whole manuscript has been revised according to the comments of all the 4 reviewers, with special attention to the unclear parts. In the sub-paragraph of Materials and Methods “4.1. Experimental Design and Statistical Analysis” the description of the experimental design has been clarified.

Round 2
Reviewer 2 Report
Comments and Suggestions for Authors
Dear authors,
Thank you very much for your sincere efforts and for addressing the issues and corrections made. The present version is fine with me.
Reviewer 4 Report
Comments and Suggestions for Authors
The authors have correctly addressed all raised issues/comments.